# Multi-Granularity Hand Action Detection

## ABSTRACT

Detecting hand actions in videos is crucial for understanding video content and has diverse real-world applications. Existing approaches often focus on whole-body actions or coarse-grained action categories, lacking fine-grained hand-action localization information. To fill this gap, we introduce the **FHA-Kitchens** (Fine-Grained Hand Actions in Kitchen Scenes) dataset, providing both coarse- and fine-grained hand action categories along with localization annotations. This dataset comprises 2,377 video clips and 30,047 frames, annotated with approximately 200k bounding boxes and 880 action categories. Evaluation of existing action detection methods on FHA-Kitchens reveals varying generalization capabilities across different granularities. To handle multi-granularity in hand actions, we propose **MG-HAD**, an End-to-End **M**ulti-**G**ranularity **H**and **A**ction **D**etection method. It incorporates two new designs: Multi-dimensional Action Queries and Coarse-Fine Contrastive Denoising. Extensive experiments demonstrate MG-HAD's effectiveness for multi-granularity hand action detection, highlighting the significance of FHA-Kitchens for future research and real-world applications. The dataset and source code will be released.

## CCS CONCEPTS

• **Computing methodologies** → **Activity recognition and understanding**.

## KEYWORDS

Hand Action Detection, Dataset, Multi-Granularity

## 1 INTRODUCTION

Action detection, a crucial task in video understanding, aims to locate and recognize action instances in each video frame, with applications in various fields [66] such as Human-Computer Interaction (HCI) [27], Smart Homes [31], the Design and Control of Robot Hands [44], and Healthcare [64]. Despite significant advancements in action recognition regarding both large-scale benchmarks [5, 54] and advanced algorithms [18, 40, 60], action detection remains relatively underexplored, mainly due to the lack of datasets with spatial action localization annotations. Moreover, existing methods predominantly focus on whole-body actions, overlooking the fine-grained actions of specific body parts, such as hands. However, hand actions are integral to daily activities, underscoring the significant research and practical importance of hand action detection.

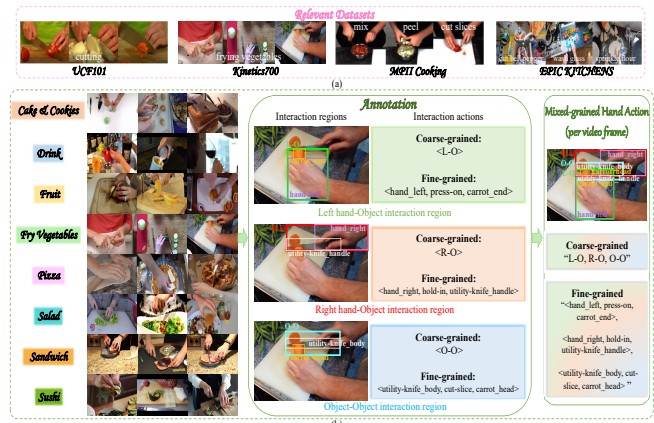

**Figure 1: Overview of the *FHA-Kitchens* dataset. (a) The annotation of hand actions in existing relevant datasets, where UCF101 [54] and Kinetics700 [53] are whole-body action datasets, while MPII Cooking [47] and EPIC KITCHENS [13] are hand action datasets. (b) The annotation of hand actions in our dataset. The left shows some frames extracted from 8 dish categories. The right illustrates the annotation process of hand actions in "*fry vegetable*".**

Pioneering datasets such as MPII Cooking Activities [47] and EPIC-KITCHENS [13] have been developed to facilitate hand-action research. However, they exhibit limitations including insufficient representation of hand-action granularity, lack of annotation of hand-action interaction regions, and neglect of the relationships between interacting objects. As shown in Figure 1(a), they offer only coarse-grained annotations for hand actions like "*cut*" rather than the fine-grained multi-dimensional categories like "*<knife, cut slice, carrot>*". These limitations hinder the study of detecting fine-grained hand actions and exploring their spatial relationship, leaving challenges in hand action detection unresolved. Therefore, establishing a large-scale benchmark with rich hand-action annotations is essential for advancing research in this field.

To this end, this paper presents a novel dataset **FHA-Kitchens**, focusing on rich and fine-grained localization and categorization information of hand actions in kitchen scenes. The FHA-Kithcens dataset encompasses a total of 2,377 video clips and 30,047 frames from eight different dish types (Figure 1(b) left). Each frame includes meticulously annotated hand action information, featuring high-quality annotations of hand interaction region boxes and corresponding coarse- and fine-grained categories. Our data were extracted from publicly available large-scale action datasets [53], focusing on videos relevant to hand actions. Subsequently, frames underwent cleaning and were annotated by ten expert voluntary annotators. To excavate more hand action information, we refined the annotation process in two aspects (Figure 1(b) right): (1) **Hand interaction regions**. These were subdivided into three

                                                                                                                                                                                                                  

sub-regions based on hand-object interaction: Left hand-Object interaction region, Right hand-Object interaction region, and Object-Object interaction region. Each sub-interaction region was annotated with bounding boxes and coarse-grained categories, denoted as "L-O", "R-O", and "O-O". (2) **Hand interaction actions**. To enhance the model's understanding of hand actions, we further refined the category of each sub-interaction region by expanding single-dimensional action categories into multi-dimensional ones, annotated in the format of triplets: *<subject, action verb, object>*, abbreviated as *<s, a, o>*, where "s-o" denotes interacting objects, and "a" represents the interaction action between the objects. Additionally, when annotating "s" and "o", we considered the specific contact area between the interacting objects, labeled as "name_contact area" (*e.g.*, "*carrot_end*"). Overall, we meticulously annotated **880** hand action categories (coarse- and fine-grained) for approximately 220k bounding boxes, with each category corresponding to a sub-interaction region's localization box. Fine-grained categories per frame have nine dimensions, resulting in 877 action triplets.

Hand Action Detection (HAD) is a sub-area of Action Detection (AD) research, which has a close relation to generic object detection (OD) in the image domain. We systematically evaluated several representative AD methods [20, 46, 65, 71] on FHA-Kitchens, observing varied performance across different levels of granularity: "Coarse-grained" and "Fine-grained". Existing methods perform significantly worse under fine-grained labels compared to coarse-grained ones, indicating that these detection methods have a better understanding of single-dimensional coarse-grained labels (*i.e.*, single verb or noun). However, real-world hand actions often involve both coarse- and fine-grained information simultaneously. Therefore, exploring the impact of multi-granularity action categories in HAD tasks is both interesting and practically significant.

Among the state-of-the-art detection methods, DINO [65] showed relatively strong performance across different granularity hand actions. Building upon DINO, we propose **MG-HAD**, a novel baseline for hand action detection (HAD). MG-HAD contains a backbone, a multi-layer Transformer encoder, a multi-layer Transformer decoder, and multiple prediction branches. To better adapt to multi-granularity hand actions, we propose two novel designs: (1) **Multi-dimensional information processing**: To enhance the model's understanding of fine-grained information, we replace the original single-dimensional content query in the decoder with multi-dimensional content queries to focus on multiple aspects of hand actions. Additionally, we introduce a Content Query Reorganization (CQR) module to generate three query sets focusing on different action dimensions as decoder inputs. (2) **Multi-granularity category processing**: We observed that the DINO's CDN (Contrastive DeNoising) module mainly focuses on bounding boxes for contrastive denoising training, while the labels are not specially designed. To enable the model to better learn and distinguish coarse-grained and fine-grained action labels, we devise coarse-grained and fine-grained sample queries for contrastive denoising training of the labels, by adding noise to different granularity categories with specified noise positions and classes. Besides, we investigate the pre-trained ResNet50 [23] and Swin-L [39] models as backbones to extract multi-scale visual features. During training, following the DN-DETR [32] method, we add ground truth labels and boxes with noises into the Transformer decoder layers to stabilize bidirectional matching, and also adopt deformable attention [71] for improved computational efficiency.

In summary, our contributions can be summarized as follows:

- To the best of our knowledge, we are the first to study the problem of multi-granularity hand action detection and establish the first hand-action dataset **FHA-Kitchens**, which includes both hand interaction region localization and multi-granularity category annotations. This dataset can serve as a benchmark for hand action detection tasks.
- We systematically investigated the impact of different granularity hand action information in kitchen scenes on the hand action detection task and provided insights about the evaluation protocol, performance analysis, and model design.
- We propose a novel multi-granularity hand action detection method named **MG-HAD**, which is designed from the perspectives of multi-granularity and multi-dimensionality. This method incorporates Multi-dimensional Action Queries and a Coarse-Fine Contrastive Denoising module to address the mixed-grained HAD problem. MG-HAD demonstrates its effectiveness in hand action detection and could serve as a strong baseline.

## 2 RELATED WORK

### 2.1 AR & AD Dataset

**Action Recognition (AR) Dataset.** Pioneering AR datasets, such as KTH [49] and Weizmann [1], have played a pivotal role in the advancement of this field, inspiring subsequent endeavors in constructing more challenging datasets. Existing studies on action recognition datasets can be divided into two main categories based on the types of actions: whole-body action and part-body action, such as UCF101 [54], Kinetics [3–5], ActivityNet [24], FineGym [50], and others [25, 28, 41, 51, 68]. These datasets primarily focus on whole-body actions, lacking fine-grained action information from specific body parts. Datasets like MPII Cooking Activities [47] and EPIC-KITCHENS [13] refine the action verb part and consider interacting objects, yet they do not describe the localization of action interaction regions or the relationships between interacting objects, crucial for HAD tasks. Representing hand actions solely with single-dimensional verbs is insufficient given the diversity and complexity of real-world scenarios. To address this issue, the FHA-Kitchens dataset enriches hand action data by providing annotations for interaction region localization and interaction action categories.

**Action Detection (AD) Dataset.** Compared to action recognition datasets, fewer datasets are available for action detection [22, 30]. This is due to the need to annotate the position and category of each action instance, which requires more efforts for dataset construction. The AVA dataset [22] focuses on human action localization, providing bounding box annotations for each person. However, this dataset primarily focuses on whole-body actions, providing location information for individuals rather than action interaction regions. Moreover, the provided action categories are mainly single-dimensional coarse-grained verbs (*e.g.*, "*sit*", "*write*", and "*stand*"). FHA-Kitchens dataset addresses these limitations by providing precise bounding box annotations for each hand sub-interaction region. Moreover, we refine the representation of action

Table 1: Comparison of relevant datasets. AR: Action Recognition. AD: Action Detection. HAD: Hand Action Detection. OD: Object Detection. ACat.: Action Category. OCat.: Object Category. Dim: Action Dimension. IRBox: Interaction Region Box.

| Dataset | Year | Ego | #Clip | Ave.Len | #Frame | #ACat. | #Verb | #OCat. | Dim | IRBox | Task |
|---|---|---|---|---|---|---|---|---|---|---|---|
| **Whole-body action dataset** | | | | | | | | | | | |
| UCF101 [54] | 2012 | × | 13.3K | ~6s | - | 101 | - | - | 1 | × | AR |
| ActivityNet [24] | 2015 | × | 28K | [5,10]m | - | 203 | - | - | 1 | × | AR |
| Kinetics400 [5] | 2017 | × | 306K | 10s | - | 400 | 359 | 318 | 2 | × | AR |
| Kinetics600 [3] | 2018 | × | 496K | 10s | - | 600 | 550 | 502 | 2 | × | AR |
| Kinetics700 [4] | 2019 | × | 650K | 10s | - | 700 | 644 | 591 | 2 | × | AR |
| AVA [22] | 2018 | × | 430 | 15m | - | 80 | 80 | 0 | 3 | × | AR,AD |
| AVA-kinetics [30] | 2020 | × | 230K | 15m,10s | - | 80 | 80 | 0 | 3 | × | AR,AD |
| FineGym [50] | 2020 | × | 32K | 10m | - | 530 | 530 | 0 | 3 | × | AR |
| **Hand action dataset** | | | | | | | | | | | |
| MPII cooking [47] | 2012 | × | 5,609 | 15m | 881K | 65 | 65 | 0 | 1 | × | AR |
| EPIC-KITCHENS [13] | 2018 | ✓ | 39.6K | 3.7±5.6s | 11.5M | 149 | 125 | 323 | 2 | × | AR,OD |
| **FHA-Kitchens** | **2024** | **✓** | **2,377** | **3m** | **30,047** | **880** | **130** | **384** | **9** | **✓** | **AR,AD,HAD,OD** |

categories and incorporate information about the interacting objects within each interaction region's action category, thereby enhancing the granularity and contextual information of hand actions. A comprehensive comparison between FHA-kitchens and existing datasets is presented in Table 1. In contrast to existing datasets, **(1)** We provide precise localization information by meticulously annotating hand interaction regions and corresponding interaction objects using bounding boxes. **(2)** We offer two granularity for hand actions: coarse- and fine-grained. For fine-grained categories, we use multi-dimensional triplets to represent each sub-interaction region action, expanding the dimensionality of each frame to 9. **(3)** We not only focus on the interacting objects that generate interaction actions but also consider the active and passive relationships between these objects, capturing their contact areas.

## 2.2 AR & AD Method

**Action Recognition (AR) Method.** Existing action recognition methods can be broadly summarized into two pipelines based on technical approaches. The first pipeline employs a 2D CNN [16, 19, 52, 61] to learn frame-level semantics and then aggregate them temporally using 1D modules. For example, TSN [60] divides an action instance into multiple segments, represents it with a sparse sampling scheme, and applies average pooling to fuse predictions from each frame. TRN [70] and TSM [34] replace pooling with temporal reasoning and shift modules, respectively. The second pipeline directly utilizes a 3D CNN [5, 15, 18, 40, 57, 62] to capture spatial-temporal semantics, such as I3D [5], SlowFast [18], and Video Swin Transformer [40]. On the other hand, AR methods can be categorized into coarse-grained [11, 12] and fine-grained [26, 37, 42, 43] based on the granularity of the actions. Some hand actions approaches [37, 42] use the EPIC-KITCHENS and MPII Cooking Activities datasets, from first-person and third-person perspectives, respectively. Another method [26] focuses on human whole-body actions in sports scenarios using the FineGym dataset [50].

**Action Detection (AD) Method.** Most state-of-the-art action detection methods [7, 17, 18, 45, 56] commonly follow a two-stage pipeline, utilizing separate 2D and 3D backbones for localization and video feature extraction, respectively. Since transformer [58] was introduced for machine translation, it has become a widely adopted

backbone for sequence-to-sequence tasks [33, 59, 65]. Most recent methods [8, 9, 21, 63, 69] utilize a unified backbone to perform action detection. VAT [21] is a transformer-style action detector designed to aggregate spatiotemporal context around target actors. EVAD [8], built upon the ViT framework, offers an end-to-end efficient video action detection method. WOO [9] and TubeR [69] are query-based action detectors that follow the detection frameworks of [2, 55] to predict bounding boxes and action classes, while STMixer [63] is a one-stage query-based detector that adaptively samples discriminative features. However, we observed that these methods primarily focus on individual human actions and overlook action interaction regions, interacting objects, and their relationships. Leveraging the advantages of transformer-based detection models, we propose an end-to-end solution capable of simultaneous hand action localization and recognition.

## 3 FHA-KITCHENS DATASET

### 3.1 Data Collection And Organization

**Data Collection.** The proposed dataset is derived from the large-scale action dataset Kinetics 700_2020 [53], which comprises approximately 650K YouTube video clips and over 700 action categories. However, as the Kinetics dataset primarily focuses on human actions, most of the videos capture whole-body actions. To narrow our focus to hand actions, we performed filtering and processing operations on the original videos in three steps. **(1)** Content Localization: We observed that videos in kitchen scenes prominently showcase human hands. So we sought out and extracted relevant videos set against a kitchen backdrop. **(2)** Quality Selection: To ensure dataset quality, we selectively chose videos with higher resolutions. Specifically, 87% of the videos were recorded at 1,280 × 720 resolution, while another 13% had a shorter side of 480. Additionally, 67% of the videos were captured at 30 frames per second (fps), and another 33% were recorded at 24~25 fps. **(3)** Duration Control: We imposed a duration constraint on the videos, ranging from 30 seconds to 5 minutes, to exclude excessively long videos. This constraint aimed to maintain a balanced distribution within the sample space. Finally, we collected a total of 2,377 video clips, amounting to 84.22 minutes of footage, encompassing 8 distinct types of dishes.

Data Organization. The collected video data was reorganized and cleaned to align with our annotation criteria (Section 3.2). First, we split the collected video data into individual frames, as our annotated units are frames. Subsequently, we conducted further cleaning of the frames by excluding those that did not depict hands or exhibited meaningless hand actions. This cleaning process took into consideration factors such as occlusion, frame quality (*i.e.*, without significant blur, subtitles, and logos), meaningful hand actions, and frame continuity. As a result, we obtained a total of 30,047 high-quality candidate video frames containing diverse hand actions for the FHA-Kitchens dataset. Compared to the initial collection, 113,436 frames were discarded during the cleaning process.

## 3.2 Data Annotation

We recruited 10 voluntary annotators to annotate hand actions for each frame with high quality. Their responsibility was to annotate bounding boxes and multi-granularity action categories for each hand interaction region. To enhance annotation efficiency, we implemented a parallel annotation approach. We utilized the LabelBee tool for annotating bounding boxes and coarse-grained categories, while fine-grained action triplets were annotated on the Amazon Mechanical Turk platform. To ensure annotation quality, we conducted three rounds of cross-checking and corrections. The annotation content and criteria are detailed below.

**Bounding Box Annotation:** We annotated the bounding boxes for both interaction regions (IR) and interaction objects (IO). (1) **IR**: We divided the hand's interaction region into three sub-interaction regions: Left hand-Object (L-O), Right hand-Object (R-O), and Object-Object (O-O) interaction regions (Figure 1(b) middle), representing regions where the left hand directly contacts an object, the right hand directly contacts an object, and objects interact with each other, respectively. The reason for focusing on O-O is that interactions between objects also involve the participation of hands. (2) **IO**: To better understand interaction actions, we also annotated the interactive object pair within each sub-interaction region using bounding boxes. For example, in L-O, we annotated objects directly touched by the left hand. In O-O, we annotated the interacting objects directly involved in hand actions (*e.g.*, *utility knife* and *carrot*). However, during annotation, we may encounter overlapping bounding boxes, *i.e.*, the same interacting object will satisfy two annotation definitions, for example, the *utility knife* in Figure 1, which is both the object directly touched by the right hand in the R-O and the active force provider in the O-O. In this case, we annotate all the labels because the same object participates in different interaction actions and has different roles (Annotation details can be seen in *supplementary material*). Finally, we annotated a total of 198,839 bounding boxes, including 49,746 hand boxes, 66,402 interaction region boxes, and 82,691 interaction object boxes.

**Hand Action Annotation:** We annotated coarse- and fine-grained actions for each sub-interaction region. Coarse-grained categories, denoted by the generic terms "L-O", "R-O", and "O-O", represent the coarse actions within the sub-interaction regions. Different from existing fine-grained datasets. We annotate each fine-grained action category in a triplet format: *<subject, action verb, object>*. (1) **Subject** & **Object**: We considered the *"active-passive"* relationship between objects, where the "subject" refers to the active force provider (*e.g.*, *utility knife*) and the "object" refers

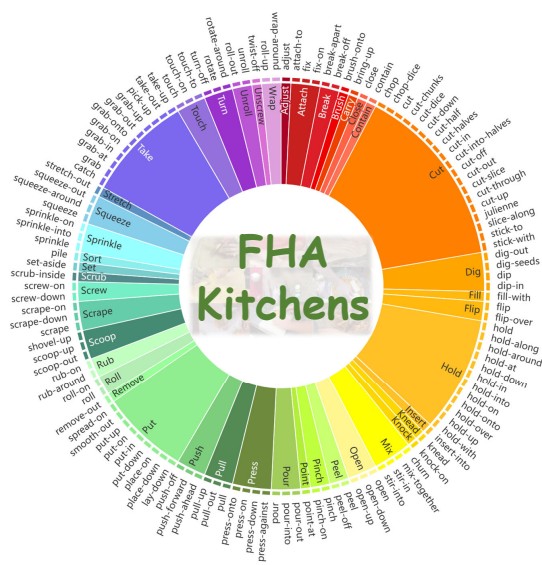

**Figure 2: An overview of the action verbs and their parent action categories in FHA-Kitchens.**

to the passive force receiver (*e.g.*, *carrot*), and annotate them in order within the action triplet. In L-O or R-O, the subject represents the corresponding hand, while the object denotes the directly interacting object. Furthermore, to enrich the description of each action, we also considered the contact areas of interacting objects within each sub-interaction region. For example, as shown in the first green block in the middle of Figure 1(b), we labeled the subject as "*hand_left*" and the object as "*carrot_end*". We referred to the EPIC-KITCHENS [13] dataset to define the object noun. (2) **Action Verb**: It describes the fine-grained hand action within the sub-interaction region. We used fine-grained verbs in the annotated action triplets and constructed the verb vocabulary by sourcing from EPIC-KITCHENS [13], AVA [22], and Kinetics 700 [4].

## 3.3 Statistics of the FHA-Kitchens Dataset

**Overview of FHA-Kitchens.** As summarized in Table 1, we annotated hand action information for 30,047 frames from 2,377 clips, resulting in 880 action categories (including 877 action triplets), 130 action verbs, and 384 interaction object nouns. We have taken steps to refine the dataset by focusing on hand action categories and interaction regions, providing more precise localization bounding boxes and rich hand action categories for the three sub-interaction regions. Compared to the original action annotations in Kinetics 700_2020 [53], the FHA-Kitchens dataset expands the action labels by 7 dimensions, increases the number of action categories by 52 times, and introduces 122 new action verbs. Furthermore, we provide bounding boxes for hand action regions (*i.e.*, 66,402 interaction region boxes). This expansion significantly enhances the diversity of hand action annotations, provides valuable region-level contextual information for each action, and facilitates future research for a wider range of video understanding tasks. The FHA-Kitchens dataset is then randomly divided into the disjoint train, validation, and test sets, with a video clip-based ratio of 7:1:2.

**Annotation Statistics.** Our annotation primarily focuses on hand interaction regions, interaction objects, and their corresponding interaction actions, resulting in a diverse array of verbs, nouns, and bounding boxes. Following the fine-grained annotation principles [13], we ensured minimal semantic overlap among action verb-noun categories, rendering them suitable for multi-category action recognition and detection. **(1) Verbs:** The annotated dataset comprises 130 action verbs that have been grouped into 43 parent verb categories (Figure 2 and Figure 3). The three most prevalent parent verb categories, based on the count of sub-action verbs, are *Cut*, *Hold*, and *Take*, representing the most frequently occurring hand actions in kitchen scenes. Figure 3 visually depicts the distribution of all verb categories within FHA-Kitchens, ensuring the presence of at least one instance for each verb category. **(2) Nouns:** In the annotation process, we identified a total of 384 interaction object noun categories that are associated with actions, categorized into 17 super-categories. Figure 4 shows the distribution of noun categories based on their affiliations with super-categories. Notably, the super-category "vegetables & plants" exhibits the highest number of sub-categories, followed by "kitchenware", which aligns with typical kitchen scenes. **(3) Bounding Boxes:** We performed a comprehensive statistical analysis on the bounding boxes of the three sub-interaction regions and the corresponding interaction objects. Specifically, we focused on two aspects: the box area and the aspect ratio. Details can be found in *supplementary material*.

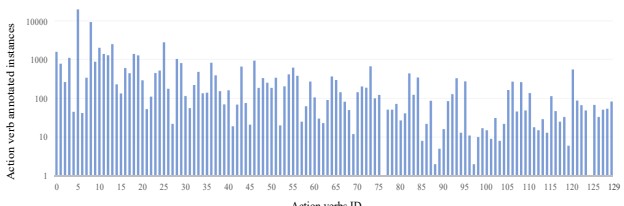

**Figure 3: The distribution of instances per action verb category (the outer ring in Figure 2) in the FHA-Kitchens dataset.**

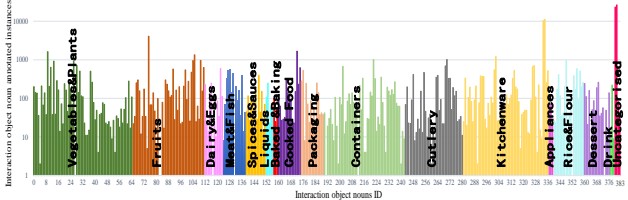

**Figure 4: The distribution of instances per object noun category from 17 super-categories in the FHA-Kitchens dataset.**

### 3.4 Benchmark Setup

**Methods.** We benchmark several representative action recognition methods [18, 40, 48, 59, 60] and detection methods [20, 46, 65, 71] with different backbone networks on the proposed FHA-Kitchens dataset based on the MMAction2 [10] and MMDetection [6] codebases. We establish three tracks using the provided dataset. **SL-AD Track**: The aim is to evaluate the supervised learning performance

**Table 2: Detection results (mAP) of hand interaction regions with different granularity levels of action categories using different methods, *i.e.*, Faster-RCNN, YOLOX, Deformable DETR, and DINO on the validation set of the SL-D track.**

| Method | Backbone | Granularity levels | |
|---|---|---|---|
| | | Coarse-Grained | Fine-Grained |
| Faster-RCNN [46] | R-50 | 65.2 | 48.5 |
| | R-101 | 66.1 | 50.0 |
| YOLOX [20] | YOLOX-s | 71.8 | 46.9 |
| | YOLOX-x | 75.6 | 49.8 |
| Deformable DETR [71] | R-50 | 73.0 | 52.4 |
| DINO [65] | R-50 | 75.2 | 53.5 |

of different detection models on hand interaction regions with different granularity levels of action categories. The results of the methods are shown in Table 2. **SL-AR Track**: This track primarily evaluates the supervised learning performance of different action recognition models on fine-grained hand actions. We trained the models with and without pre-trained weights on the FHA-Kitchens dataset. **DG Track**: It focuses on experiments for Intra- and Inter-class Domain Generalization in Interaction Region Detection, exploring both intra-class and inter-class perspectives. All models on the SL-AD, SL-AR, and DG tracks were trained and tested using NVIDIA GeForce RTX 3090 GPUs. For the SL-AD and DG tracks, we employ the mean Average Precision (mAP) [36] as the primary evaluation metric, while for the SL-AR track, Top-1 accuracy and Top-5 accuracy (%) are adopted. Detailed results of SL-AR and DG can be found in *supplementary material*.

**Results and Discussion.** The results in Table 2 show that current detection methods perform well in learning single-dimensional coarse-grained categories like verbs or nouns. However, they struggle in learning multi-dimensional fine-grained action categories. Understanding the intricate nature of real-world hand actions, which encompass both coarse- and fine-grained information, underscores the significance of investigating multi-granularity action categories in HAD tasks, an area that poses significant challenges and remains largely unexplored. To fill this gap, we propose a novel method for multi-granularity hand action detection.

## 4 A SIMPLE YET STRONG BASELINE

### 4.1 A Multi-Granularity Framework

Drawing inspiration from the image-based DINO [65], we propose the novel MG-HAD method with specific novel designs in the decoder for multi-granularity hand action detection (Figure 5). MG-HAD consists of a backbone, a multi-layer Transformer encoder, a multi-layer Transformer decoder, and multiple prediction branch heads. Given a video clip, for each frame, we utilize backbones like ResNet [23] or Swin Transformer [39] to extract multi-scale features, which are then fed into the Transformer encoder along with corresponding positional embeddings. After enhancing features through the encoder layers, we initialize anchors as positional queries for the decoder using a mixed query selection strategy, following the design of DINO, without initializing content queries but leaving them learnable. It's worth noting that the original content

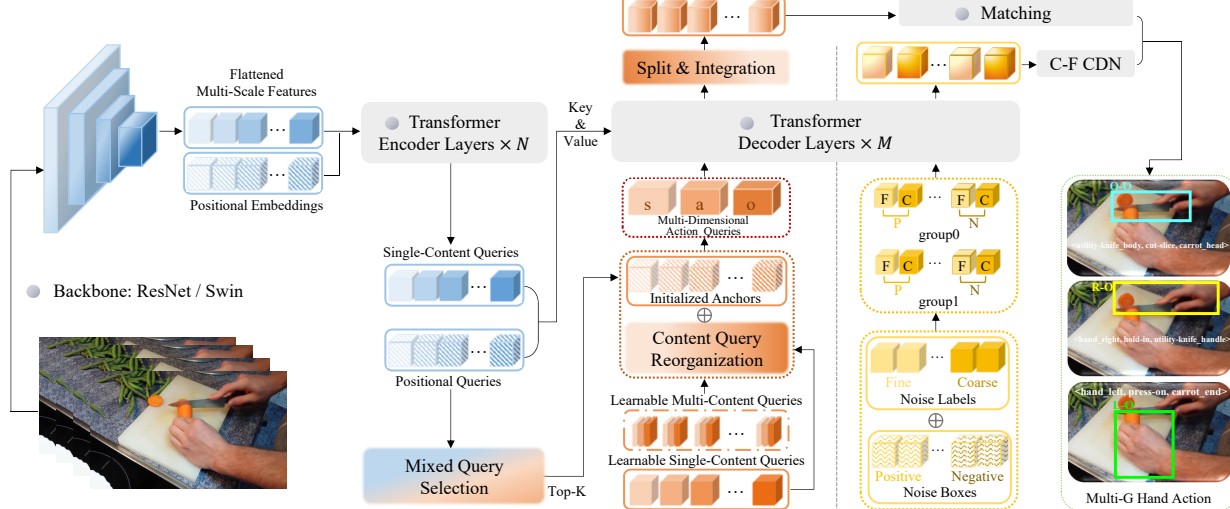

**Figure 5: The overall architecture of MG-HAD, a novel end-to-end hand action detection model based on DINO [65]. The improvements mainly focus on the decoder part. Specifically, (1) we introduce a new design for the content query part, transforming the original single-dimensional content queries into multi-dimensional ones. They are further processed by the designed CQR module, combined with initialized anchors, and inputted into the decoder. The outputted three query sets with different action dimensions go through the Split & Integration module to generate $N$ queries containing three action dimensions. Finally, the matching process is conducted to predict hand action results (see Section 4.2); (2) we introduce a C-F CDN training approach, which involves adding coarse- and fine-grained noise to labels to generate four types of CDN queries for contrastive denoising training (see Section 4.3). F: Fine-grained, C: Coarse-grained, Multi-G: Multi-granularity.**

queries focus on semantic information of single-dimensional categories, which is not suitable for fine-grained multi-dimensional categories in the new task. Therefore, we modify the single-content query to multi-content queries and introduce a Content Query Reorganization (CQR) module to obtain query sets focusing on three different sub-action dimensions, as detailed in Section 4.2. Additionally, similar to DINO, we have an extra CDN branch to perform contrastive denoising training. In contrast to the standard CDN method, we specifically devise a novel coarse-fine granularity contrastive denoising training approach to distinguish labels with different granularity levels, which will be discussed in Section 4.3.

### 4.2 Multi-Dimensional Action Queries

Comparing the results presented in Table 2, it's clear that existing detection methods struggle with learning from multi-dimensional fine-grained labels. Fine-grained hand action detection poses a greater challenge compared to coarse-grained detection due to the need to discern subtle differences in similar hand actions. Additionally, multi-dimensional fine-grained labels provide important supervisory signals about subject, object, and action categories, as well as localization information. However, effectively encoding this information at different dimensions and leveraging these supervisory signals, particularly in terms of query design within the DETR series framework [2, 65, 71], remains unexplored.

**Implementation:** We observed that the current design of content queries mainly focuses on single-dimensional semantic information, i.e., single verb or noun categories. However, in fine-grained categories, we incorporate both verb and noun categories, generating multi-dimensional semantic information, i.e., <$c_1, c_2, c_3$> ($c_1, c_3$

$\in$ *nouns*, and $c_2 \in$ *verbs*), or more specific <*s, a, o*>. If we stick to the original design, content queries would consider <*s, a, o*> as a whole, learning global information from a single-dimensional perspective. To enhance the model's focus on local information of sub-categories, we transform a set of content queries $Q = \{q_1, ..., q_n\}$ originally focusing on single dimensions into three sets of content queries, i.e., $Q_s$, $Q_a$, and $Q_o$, focusing on different action dimensions. $n$ is the index of the original queries. Specifically, we first convert each query element $q_n$ (*bottom orange cubes in Figure 5*) into three sub-queries, i.e., $q_{n_s}$, $q_{n_a}$, and $q_{n_o}$, expanding $N$ original queries to $3 \times N$ sub-queries. Next, through our designed Content Query Reorganization (CQR) module, sub-queries focusing on the same action dimension (i.e., $q_{1_s}, q_{2_s}, ..., q_{n_s}$) are selected and reorganized to obtain a query set for each action dimension. Additionally, to ensure a comprehensive understanding of fine-grained categories, we introduce an action dimensional hyper-parameter $w_d$ ($d \in \{s, a, o\}$), to add a certain proportion of weight to each query set, which is then sum with the global information (i.e., $Q = \{q_1, ..., q_n\}$). This process is formulated as:

$$Q_d = CQR\left(\{q_{n_d}\}_{n=1}^{N}\right) = \sum_{n=1}^{N} q_{n_d} \times w_d + Q, \ d = s, a, or \ o. \quad (1)$$

After passing through the CQR module, we obtain content query sets for the three action dimensions. These sets are then summed with the initialized anchors to yield multi-dimensional action queries. Each dimensional query set has a length of $N$, resulting in a total length of $3N$. Following the decoder layers, three query sets for different action dimensions ($Q'_s$, $Q'_a$, and $Q'_o$) are outputted.

Subsequently, through the Split&Integration module, queries from different action dimensions with the same index (*e.g.*, $q_{1_s}$, $q_{1_a}$, $q_{1_o}$) are integrated to generate $N$ queries, each of which contains information from three action dimensions (*top orange cubes in Figure 5*):

$$\{q_n\}_{n=1}^N = SI\left(Q_s', Q_a', Q_o'\right) = \left\{q_{n_s} + q_{n_a} + q_{n_o}\right\}_{n=1}^N, \quad (2)$$

where *SI* represents the Split&Integration module. Finally, the matching process is conducted to predict hand action results.

**Analysis:** In our design of multi-dimensional action queries, we introduce an action dimensional hyper-parameter $w_d$ ($d \in \{s, a, o\}$), to control the proportion of local information (sub-categories) fused with global information (triplet categories). In the three action dimensions <*s, a, o*>, the *a* dimension is the most crucial for our task. Therefore, we use $w_a$ as the central weight to dynamically adjust the weight proportions of the three action dimensions, with a total sum of 1. To determine the optimal weight distribution, we conducted a total of 10 comparative experiments under different backbones, with detailed results provided in *supplement material*.

### 4.3 Coarse-Fine Contrastive Denoising

For object detection, DINO is highly effective in stabilizing training and accelerating convergence. With the help of DN queries, it learns to predict "no object" for anchors without nearby objects, thereby inhibiting confusion and selecting high-quality anchors (queries) for predicting bounding boxes. However, in HAD tasks where hand action categories may overlap or be similar, DINO primarily addresses the confusion of boxes but overlooks label categories, resulting in poor prediction capability for different granularity levels of hand action categories. To address this issue, we propose a **C**oarse-**F**ine granularity **C**ontrastive **DeN**oising (C-F CDN) training approach to reject anchors with "incorrect granularity labels".

**Implementation:** DINO introduces two hyper-parameters $\gamma_1$ and $\gamma_2$, to control the scale of box and label noise, respectively. The generated noises are no larger than $\gamma_1$ and $\gamma_2$, aiming to enable the model to reconstruct the ground truth (GT) from moderately noisy queries. We observed that DINO only designs two types of CDN queries for the box: positive and negative queries, while the label is set to be randomly generated. In the proposed method, while keeping the box settings unchanged, we further generate two types of CDN queries for the label: coarse-grained and fine-grained queries (*dark and light yellow cubes in Figure 5*). Moreover, unlike the strategy of randomly generating noisy labels, we add noise by specifying the noise position and noise category for different granularity labels. Specifically, coarse-grained queries add noise containing fine-grained information, while fine-grained queries add noise containing coarse-grained information, with the expectation of predicting the correct granularity label for each GT box. In Figure 5, each CDN group comprises four types of queries: positive-coarse, positive-fine, negative-coarse, and negative-fine. If a frame has $n$ GT bounding boxes, a CDN group will contain 4 types of $2 \times n$ queries. Similar to DINO, we also utilize multiple CDN groups to enhance the effectiveness of the method. The reconstruction loss for bounding box regression includes $l_1$ and GIOU losses, while focal loss [35] is employed for classification.

**Analysis:** When designing the noise label generation strategy, we replaced the "random" generation of noise with "specified", reducing randomness by specifying noise positions and categories.

**Table 3: Results for MG-HAD and other DETR Series detection models with the ResNet50 backbone on the FHA-Kitchens validation set trained with *12 epochs*. M-G: Mixed-Grained, C-G: Coarse-Grained, F-G: Fine-Grained.**

| Method | FHA-Kitchens val mAP(%) | | |
|---|---|---|---|
| | M-G label | C-G sub-label | F-G sub-label |
| *DETR* [2] | 42.3 | 72.8 | 41.9 |
| *Deformable DETR* [71] | 49.4 | 70.9 | 49.1 |
| *DAB-DETR* [38] | 52.1 | 73.1 | 51.8 |
| *DDQ-4scale* [67] | 53.8 | 67.8 | 53.7 |
| *DINO-4scale* [65] | 54.7 | 76.3 | 54.5 |
| ***MG-HAD-4sacle*** | **57.0(+2.3)** | **75.6** | **56.8(+2.3)** |

This ensures noise is added to different granularity labels, generating CDN queries encompassing various granularity. To determine the optimal setting, we considered the noise distribution of different granularity categories in real-world scenarios and ensured contrastive learning between coarse and fine-grained information. We conducted three sets of comparative experiments (Details can be found in *supplementary material*). The final selected setting, as shown in Eq. (3), exhibits the most significant improvement. Hence, subsequent experiments were conducted using this setting for further investigation. Our method's success lies in its ability to suppress confusion at the category level and select appropriate granularity to predict hand action categories, thus enhancing its ability to predict multi-granularity information.

$$\text{noise label} = \begin{cases} \text{fine-grained} & i \in [0, 3) \\ \text{mixed-grained} & i \in [3, C) \end{cases}, \quad (3)$$

where $i$ indexes the multi-granularity action category for a specific instance (*i.e.*, 0~2 denote coarse-grained categories while 3~C-1 denote fine-grained categories), and $C$ is the number of categories. "fine-grained" and "mixed-grained" denote that the noise label is chosen randomly from the fine-grained categories and the combination of the coarse-grained and fine-grained categories, respectively.

## 5 EXPERIMENTS

### 5.1 Experiments Settings

**Dataset and Metric.** Due to the absence of benchmarks for this new task, we evaluated our model and other representative detection models solely on the FHA-Kitchens dataset. We conducted experiments using two different backbones: ResNet-50 [23] pre-trained on ImageNet-1k [14] and Swin-L [39] pre-trained on ImageNet-22k [14]. All detection models utilize pre-trained weights on the MS COCO object detection dataset [36]. Furthermore, we not only report the overall validation results using mixed-grained labels but also separately report the validation results for coarse-grained and fine-grained sub-labels. We follow previous works and adopt mean Average Precision (mAP) [36] as the primary evaluation metric.

**Implementation Details.** We trained the MG-HAD model on the FHA-Kitchen dataset using the MMDetection [6] codebase. Specifically, we utilized pre-trained weights on the MS COCO [36]

**Table 4: Results of MG-HAD and other SOTA detection models on the FHA-Kitchens validation set. R-50: ResNet-50, M-G: Mixed-Grained, C-G: Coarse-Grained, F-G: Fine-Grained.**

| Method | Epoch | Backbone | FHA-Kitchens val mAP(%) | | |
|---|---|---|---|---|---|
| | | | M-G label | C-G sub-label | F-G sub-label |
| *Faster R-CNN* [46] | 108 | R-50 | 48.3 | 22.3 | 48.6 |
| *YOLOX* [20] | 100 | YOLOX-x | 50.7 | 70.8 | 50.5 |
| *DETR* [2] | 150 | R-50 | 50.6 | 73.1 | 50.3 |
| *Deformable DETR* [71] | 50 | R-50 | 53.7 | 72.6 | 53.4 |
| *DAB-DETR* [38] | 50 | R-50 | 54.7 | 75.2 | 54.5 |
| *DINO-4scale* [65] | 24 | R-50 | 56.3 | 74.5 | 56.0 |
| *DINO-4scale* [65] | 12 | R-50 | 54.7 | 76.3 | 54.5 |
| *DINO-5scale* [65] | 12 | Swin-L | 56.3 | 76.3 | 56.1 |
| **MG-HAD-4scale** | 24 | R-50 | **57.7**(+1.4) | **75.3** | **57.5**(+1.5) |
| | 12 | | **57.0**(+2.3) | **75.6** | **56.8**(+2.3) |
| **MG-HAD-5scale** | 12 | Swin-L | **59.4**(+3.1) | **77.6** | **59.2**(+3.1) |

object detection dataset and fine-tuned it on the hand action detection task on FHA-Kitchens. We trained the model under two different settings: 4scale-R-50 and 5scale-Swin-L. Following DINO [65], we used the Adam optimizer [29] for model training, with an initial learning rate of $1 \times 10^{-4}$ and weight decay is $10^{-4}$. The experiments were conducted on the NVIDIA GeForce RTX 3090 GPUs, with a batch size of 2 for 4scale-R-50 and 1 for 5scale-Swin-L. By default, MG-HAD was trained for 12 epochs, taking approximately 5 hours. More details are provided in the *supplement material*.

## 5.2 Main Results

**12-Epoch Setting.** To demonstrate the effectiveness of our method for the multi-granularity HAD task, we compared it with representative strong baselines from the DETR series [2, 38, 65, 67, 71] on the FHA-Kitchens dataset under the setting of ResNet-50 backbone and 12 epochs. In particular, our method, DINO [65], and DDQ [67] mainly report results under the 4scale setting. As shown in Table 3, our method achieves much better accuracy in detecting mixed-grained hand actions, owing to the proposed C-F CDN module and multi-dimensional action queries. Specifically, it achieves an improvement of *+2.3* AP on mixed-grained labels compared to the current strongest baseline DINO [65] under the same setting. Furthermore, compared to the classic DETR [2], our method achieves a significant improvement of *+14.7* AP. Note that our method not only performs well for mixed-grained labels but also shows improvement in the validation results for fine-grained sub-labels.

**Comparison with SOTA Detection Methods.** To comprehensively and fairly validate the effectiveness of our method in enhancing the performance of multi-granularity hand actions, we compared it with other state-of-the-art (SOTA) detection methods on the FHA-Kitchens dataset, utilizing their optimal settings (refer to the MMDetection [6] codebase). DINO exhibits relatively fast convergence, achieving good results with just 12 epochs on the Swin-L backbone. Our method inherits the convergence capability of DINO but yields more significant improvements. We adopted the same settings as DINO [65], utilizing both 4scale ResNet-50 and 5scale Swin-L backbones, trained for 12 epochs and 24 (2×) epochs, respectively. The results in Table 4 indicate the following: **(1)** Our method exhibits a significant improvement compared to the baseline [65], which can be attributed to the design of handling fine-grained information in our model; **(2)** Comparing models trained

**Table 5: Ablation study of the key components in MG-HAD. C-F CDN: Coarse-Fine granularity Contrastive De-Noising Training, Multi-DA Q: Multi-Dimensional Action Queries.**

| Method | Algorithm Components | | mAP(%) | |
|---|---|---|---|---|
| | *C-F CDN* | *Multi-DA Q* | *4scale-R-50* | *5scale-Swin-L* |
| **Baseline** [65] | | | 54.7 | 56.3 |
| | ✓ | | 56.6 | 58.2 |
| | | ✓ | 56.4 | 58.7 |
| **MG-HAD** | ✓ | ✓ | **57.0** | **59.4** |

for 24 epochs and 12 epochs, the main improvement lies in the accuracy of fine-grained action detection. Since the FHA-Kitchens dataset contains overwhelming fine-grained categories over the coarse-grained ones, the model's representation capacity may be primarily utilized for fitting fine-grained categories; **(3)** Under the 5-scale Swin-L backbone, our method achieves a significant improvement of *59.4* AP for mixed-grained hand actions with just 12 epochs. This indicates that using a more powerful backbone [39] can improve both coarse- and fine-grained action detection accuracy. Specifically, the detection accuracy of fine-grained actions is increased by *+3.1* AP and the detection accuracy of coarse-grained actions is increased by *+1.3* AP. The visualization of the detection results can be found in the *supplement material*.

## 5.3 Ablation Studies

**Effectiveness of New Components:** Our method utilizes the multi-dimensional action queries for multi-dimensional information processing, as introduced in Section 4.2, and the C-F CDN module for multi-granularity information processing, as described in Section 4.3. To further validate the effectiveness of these components, we separately isolated them from the model and evaluated the performance under two settings: 4scale ResNet-50 and 5scale Swin-L, as shown in Table 5, where the baseline denotes the original design proposed by DINO [65]. As can be seen, while the strong baseline DINO [65] has already surpassed previous models, the proposed MG-HAD introduces two novel designs that notably boost performance in hand action detection. Each module significantly enhances the baseline on both backbones, and their combined effect further enhances performance, demonstrating their complementary role in understanding multi-granularity hand action information.

## 6 CONCLUSION

In this paper, we present the first study on multi-granularity hand action detection, aiming to understand the diverse hand actions through localizing regions and recognizing various granularity categories of hand actions. We establish **FHA-Kitchens**, the first fine-grained hand action detection dataset, comprising 30,047 high-quality video frames, 198,839 bounding boxes, and 880 hand action categories. Through systematic evaluation, we identify that existing detection methods excel in coarse-grained actions but struggle with fine-grained ones. To address this, we propose **MG-HAD**, a simple yet strong baseline model leveraging the Transformer detector with two novel designs. It outperforms previous methods across various granularities of hand actions. FHA-Kitchens and MG-HAD can serve as a valuable testbed and baseline for future research.

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
