# OpenReview forum: "Multi-Granularity Hand Action Detection"
_acmmm.org/ACMMM/2024/Conference — MM2024 Poster_

### Official Review · Reviewer_MLGM · 2024-05-23

**Rating:** 4
**Confidence:** 4

**Summary:**

The core contribution of this paper is the multi-granularity hand action detection dataset named FHA-Kitchens, which provides fine-grained hand action localization and classification information. The dataset fills the gaps in action localization and fine-grained classification in previous full-body action datasets and hand action datasets. To better handle multi-granularity hand actions, the authors propose the MG-HAD method, which introduces multi-dimensional action queries and a coarse-fine contrastive denoising module.

**Strengths:**

- This paper is the first to study the problem of multi-granularity hand action detection. The proposed FHA-Kitchens dataset provides coarse-grained action category annotations while also offering fine-grained multi-dimensional annotations.

- The annotation work of the dataset was completed by 10 expert annotators and underwent three rounds of cross-checking to ensure annotation quality. The action verbs and interacting object nouns covered in the dataset are broad and well-distributed.

- The MG-HAD method based on the DINO architecture, effectively solves the problem of multi-granularity hand action detection.

- The paper is well written.

**Limitations:**

- In addition to the interactions of "left hand-object", "right hand-object", and "object-object", should the interaction of "left hand-right hand" also be considered?

- For the macro task of "hand action detection", the dataset proposed in this paper lacks annotations of hand pose, so the description of "hand action" tends to focus on the interaction between hands and objects (action triplets), rather than hand actions in the traditional sense.

- How does the MG-HAD method perform on traditional datasets (such as UCF101, Kinetics700, EPIC-Kitchens) in terms of action recognition or action detection? Does the MG-HAD method of introducing multi-granularity hand actions exhibit fine generalization?

**Suitability:**

2

---

### Official Review · Reviewer_wFdR · 2024-05-24

**Rating:** 5
**Confidence:** 3

**Summary:**

This article proposes a new video action recognition dataset for kitchen hand action interaction, which includes multi granularity interaction actions and detection boxes, filling a certain gap in the field of video action recognition. In addition, to address the multi granularity problem in hand action recognition tasks, the paper also proposes MG-HAD, an end-to-end multi granularity hand action detection method.

**Strengths:**

The article proposes a new dataset for human action recognition that includes multi granularity annotations. This dataset includes multiple annotation forms and can be applied to various tasks including action detection, action recognition, hand action recognition, object detection, action interaction recognition, etc. It provides a data foundation for subsequent research on multi granularity action recognition. In addition, the article also provides a baseline for solving multi granularity hand action recognition for comparison.

**Limitations:**

The number of videos and total frame rates in the new dataset are relatively small, and future practical research in application may be limited. In addition, the other models provided in the paper for comparative testing are not novel enough and are relatively old.

**Suitability:**

3

---

### Official Review · Reviewer_v3aD · 2024-06-01

**Rating:** 4
**Confidence:** 3

**Summary:**

The paper focuses on the challenge of detecting fine-grained hand actions in videos, specifically within kitchen scenes, which has diverse real-world applications. To address this, the authors introduce the FHA-Kitchens dataset, comprising 2,377 video clips and 30,047 frames annotated with approximately 200k bounding boxes and 880 action categories. This dataset offers detailed annotations for both coarse- and fine-grained hand actions, filling a significant gap in the field.

Furthermore, the paper presents MG-HAD (Multi-Granularity Hand Action Detection), an innovative end-to-end method designed to handle multi-granularity hand actions. MG-HAD incorporates Multi-dimensional Action Queries and Coarse-Fine Contrastive Denoising to enhance detection accuracy. Extensive experiments demonstrate MG-HAD's effectiveness, highlighting the importance of the FHA-Kitchens dataset for future research and practical applications. The results underscore the method's superior performance in detecting hand actions across different granularity levels, establishing a new benchmark in the field.

**Strengths:**

Novelty: Introducing the FHA-Kitchens dataset, which provides comprehensive annotations for both coarse- and fine-grained hand actions, filling a critical gap in the field.

Technical Approach: The MG-HAD method is robust, incorporating Multi-dimensional Action Queries and Coarse-Fine Contrastive Denoising to effectively handle multi-granularity hand actions.

Evaluation: Extensive experiments demonstrate MG-HAD's superior performance on the FHA-Kitchens dataset, providing a detailed comparison with existing methods.

**Limitations:**

Lack of Novelty in Methodology: While the FHA-Kitchens dataset is novel, the proposed MG-HAD may not present significant innovation compared to existing methods. The paper could benefit from a clearer articulation of the unique challenges addressed by their approach.

Dataset Scope: Although FHA-Kitchens is detailed, it is limited to kitchen scenes. This narrow focus may restrict the generalizability of the findings to other contexts or environments where hand actions are critical.

Clarity on Annotation Process: While the annotation process is described, more details on the consistency and accuracy of the annotations would be beneficial. This includes how annotator bias was minimized and how inter-annotator agreement was ensured.

**Suitability:**

3

---

### Meta-Review · Area_Chair_xGTD · 2024-06-30

**Recommendation:** Accept (Poster)
**Confidence:** 5

**Metareview:**

This work is focused on detecting fine-grained hand action in videos. It proposed a new dataset for this task with detailed bench-marking/comparison of existing methods. It initially received 2x borderline accept and weak accept ratings. The main concerns were limited size of dataset, limited scope to kitchen settings, and weak comparisons. The authors provide a rebuttal and most of the concerns were addressed. The final ratings are 2x weak accept and borderline accept. The most critical remaining concerns are size of dataset and used baselines. These limitations are important, but the other strengths of this work makes this a good contribution to the community and AC recommends acceptance of this work.